# The Role of A20 in Cancer: Friend or Foe?

**DOI:** 10.3390/cells14070544

**Published:** 2025-04-04

**Authors:** Jinju Lee, Heesun Cheong

**Affiliations:** Division of Cancer Biology, Research Institute, National Cancer Center, Goyang-si 10408, Republic of Korea; jj3168@ncc.re.kr

**Keywords:** A20/TNFAIP3, cancer, autophagy

## Abstract

A20 is a ubiquitin-editing enzyme that has emerged as a key regulator of inflammatory signaling with paradoxical roles in cancer. Acting as both an oncogene and a tumor suppressor gene depending on the cellular context, A20 modulates important cell pathways, such as NF-κB signaling and autophagy. In this review, we summarize the dual roles of A20 in tumorigenesis, highlighting its ability to promote tumor progression in cancers, such as breast and melanoma, while functioning as a tumor suppressor in lymphomas and hepatocellular carcinoma. We discuss the interplay of A20 with autophagy, a process that is important for maintaining cellular homeostasis and influencing tumor dynamics. By integrating recent findings, we provide insight into how dysregulation of A20 and its associated pathways can either suppress or drive cancer development, which may lead to improved therapeutic intervention.

## 1. Introduction

A20, also known as TNFAIP3 (Tumor Necrosis Factor Alpha-Induced Protein 3), is an important protein that controls inflammation and immunity. It is a ubiquitin (Ub)-editing enzyme characterized by two distinct Ub-editing domains, which facilitate its multifunctional regulatory capacity. The C-terminus of A20 contains seven ZnF domains and an N-terminal ovarian tumor (OTU) domain. The ZnF4 domain exhibits selective binding activity for the K63-linked polyubiquitin in vitro system [1], whereas the ZnF7 domain functions as a ubiquitin-binding domain with high affinity for M1-linked Ub [2,3]. In contrast, the N-terminal OUT domain acts as a deubiquitinating domain (DUB) that facilitates the removal of ubiquitin chains, which generates the dynamic versatility of A20 in cell signaling [4].

A20 was originally identified as a downstream effector of the tumor necrosis factor (TNF) receptor. It gained recognition for its ability to inhibit TNF-induced cell death [5]. Further studies revealed that A20 is a negative regulator of TNF-induced NF-κB activation, a pathway involved in various biological processes, including inflammation, immune responses, and cell survival [6,7,8]. By fine-tuning NF-κB activation, A20 plays an important role in maintaining immune homeostasis and preventing an excessive inflammatory response. Accordingly, A20 has been considered a key modulator in the initiation and progression of autoimmune diseases, in which it is dysregulated by pro-inflammatory NF-κB signaling [6,7,8].

Because of the well-established link between chronic inflammation and cancer, A20 has gained attention for its involvement in tumor biology. Studies suggest that the dysregulation of A20 contributes to various cancers; however, the precise role of A20 in cancer remains incompletely understood, as its functions vary depending on the cellular context and specific signaling pathways. In some cancers, A20 acts as a tumor suppressor by attenuating pro-inflammatory and pro-survival signals, whereas in others, it facilitates tumor progression through its effects on cell survival and immune evasion [9,10,11].

In this review, we summarize recent reports that explore the dual roles of A20 in cancer, focusing on its complex involvement in tumor progression. We also discuss the emerging evidence linking A20 to autophagy, highlighting its potential role as a modulator of cancer development through this important cellular process.

## 2. A20, a Regulator of Inflammation

A20 plays an important role as a regulator of inflammation and immune homeostasis. Its primary role is to control the NF-κB signaling pathway, which serves as a master regulator of inflammatory and immune responses [6,7,8]. The NF-κB pathway is activated by various stimuli, including pro-inflammatory cytokines, such as tumor necrosis factor-alpha (TNF-α), interleukin-1β (IL-1β), microbial products like lipopolysaccharides (LPSs), and signals from pattern recognition receptors, such as Toll-like receptors (TLRs) [6,7,8]. Upon activation, NF-κB translocates to the nucleus, where it induces the transcription of pro-inflammatory genes involved in cytokine production, cell survival, and immune activation. To prevent excessive or prolonged activation, A20 also acts as a negative feedback regulator, ensuring that NF-κB signaling is transient and tightly controlled [6,7,8].

A20 regulates NF-κB signaling through its dual ubiquitin-editing enzymatic activities. Both the OTU and ZnF4 domains are essential for A20 function, as mutations in either domain lead to stabilization of its substrate, receptor-interacting protein 1 (RIP1) [4]. Specifically, A20 removes K63-linked polyubiquitin chains from key signaling intermediates, such as RIP1, thereby inhibiting TNF-α-induced NF-κB signaling. By deubiquitinating RIP1, A20 prevents the formation of signaling complexes that are necessary for sustained NF-κB activation. Simultaneously, A20 also functions as an E3 ubiquitin ligase, promoting K48-linked polyubiquitin chains to RIP1 and other substrates in cooperation with ITCH, RNF11, and TAXBP-1, which mark these signaling proteins for proteasomal degradation, and further dampening inflammatory responses [12,13].

This dual enzyme activity not only disrupts NF-κB signaling, but also clears signaling intermediates, which effectively terminates the inflammatory response [4,7,14]. In addition to its role in TNF-α-induced signaling, A20 regulates NF-κB activation in response to signaling downstream of other receptors, including TLRs, IL-1 receptors (IL-1Rs), and NOD-like receptors. For example, in TLR signaling, A20 deubiquitinates and degrades intermediates, such as TRAF6 and TAK1, which are required for NF-κB activation [4,7,14]. This action prevents overactivation of innate immune responses to microbial components and limits the production of pro-inflammatory cytokines.

The loss or dysfunction of A20 results in uncontrolled NF-κB activation, which drives hyper-inflammation and autoimmunity. Studies in A20-deficient mice revealed severe systemic inflammation, multi-organ damage, and premature death because of the unchecked activity of NF-κB-dependent pro-inflammatory genes [4,15,16,17,18,19,20]. In humans, mutations or reduced expression of A20 are associated with autoimmune and inflammatory diseases, including rheumatoid arthritis [21,22], systemic lupus erythematosus [23,24], inflammatory bowel disease [20], and psoriasis [25]. Therefore, by ensuring the inactivation of NF-κB, A20 prevents excessive immune responses, maintains immune homeostasis, and protects against inflammation-driven diseases, which highlights its important role in maintaining health and preventing disease.

## 3. Oncogenic Role of A20 in Cancer

Although the role of A20 in various cancers is multifaceted, studies suggest that A20 predominantly acts as an oncogene in certain tumor types to promote tumor progression, metastasis, and drug resistance (Figure 1).

In breast cancer, A20 is highly expressed and is associated with tumor growth, metastasis, and poor prognosis [26,27,28,29]. Moreover, high A20 expression levels in breast cancer patients are closely associated with negative recurrence-free survival and overall survival ratios, as well as a poor prognosis [30]. Regarding its mechanism, several studies have suggested that increased expression of A20 in breast cancer suppresses TNFα-induced apoptosis, thereby promoting tumor growth and metastasis [27,28].

In triple-negative breast cancer (TNBC), TNFα exposure upregulates A20 expression, which promotes tumor growth and invasion, as evidenced in mouse xenograft models. A20 enhances cancer metastasis by directly interacting with Snail1, a key regulator of EMT, which is important for breast cancer cell invasion and metastasis. A20 directly binds and mediates the multi-monoubiquitination of Snail1, stabilizing the protein and protecting it from proteasomal degradation. This post-translational modification of Snail1 enhances its activity, promoting EMT and facilitating the metastatic potential of basal-like breast cancer cells [27]. In contrast, luminal (ER+) breast cancer cells exhibit significantly increased sensitivity to TNFα-induced cell death in the absence of A20, which was prevented by A20 overexpression via activating the HSP70-mediated anti-apoptotic pathway [28]. A20 also promotes epithelial–mesenchymal transition (EMT), a cancer stem cell phenotype, and increased metastatic potential by activating pStat3 signaling. In mouse models, A20 overexpression in luminal cancer cells also recruits granulocytic myeloid-derived suppressor cells (MDSCs), contributing to a tumor-promoting inflammatory microenvironment [28]. These findings highlight a novel mechanism by which A20 contributes to the progression and aggressiveness of this breast cancer subtype (Figure 1A,C).

In melanoma and gastric cancer, the expression of A20 is also markedly increased. A20 promotes melanoma cell proliferation, survival, and metastasis by activating the Akt signaling pathway [31]. Through its ubiquitin-editing functions, A20 stabilizes and activates key components of the Akt pathway, resulting in increased phosphorylation and activation of Akt. This activation promotes cell survival, growth, and motility, which contributes to the aggressive behavior of melanoma cells. Moreover, a recent study highlighted that the regulatory effects of A20 on inflammation and immune responses may create a microenvironment conducive to tumor growth. Indeed, high levels of A20 in tumor cells correlate with a poor response to anti-PD-1 therapy [10,11]. A20 promotes immune evasion by regulating PD-L1 expression and decreasing the infiltration and activity of CD8^+^ T cells. Decreasing A20 expression in tumors enhanced the efficacy of anti-PD-1 therapy in mice, suggesting that targeting A20 could improve immunotherapy outcomes in melanoma patients.

In gastric cancer, studies have shown controversial roles for A20, although its expression level is highly increased [9,32]. One study identified an oncogenic role: hypomethylation of promoter regions in the NRN1 and A20 genes results in their increased expression, with a correlation with more aggressive tumor behavior and poorer prognosis [33]. However, under specific conditions, such as Helicobacter pylori infection, A20 acts as a tumor suppressor gene by inhibiting the NF-κB pathway and reducing the expression of anti-apoptotic genes [34]. The suppression of alternative NF-κB activity by A20 results in increased apoptosis of infected gastric epithelial cells, which may limit the survival and proliferation of H. pylori-infected cells. This dual role highlights the complex and context-dependent nature of A20 in gastric cancer [34].

The role of A20 in colorectal cancer is similarly complex. A20 has been recognized as a tumor suppressor, with studies reporting reduced expression resulting from genetic and epigenetic alterations [35,36,37,38]. However, a recent study indicates an oncogenic role for A20 in a certain context of colorectal cancer. For example, increased A20 expression reduces calreticulin (CRT) on the tumor cell surface, thereby promoting tumor growth and immune evasion through a stanniocalcin 1 (STC1)-CRT–dependent pathway [39] (Figure 1B). Moreover, high A20 expression levels are associated with a poor response to PD-1 blockade treatment and reduced survival rates, indicating its role in immune resistance and tumor growth.

The role of A20 in pancreatic cancer has not been clearly revealed yet. However, a recent report showed its oncogenic role through TRAIL (TNF-related apoptosis-inducing ligand)-induced apoptosis. A20 has also been implicated in resistance to TRAIL-induced apoptosis. TRAIL is a promising therapeutic agent that selectively induces apoptosis in cancer cells, while sparing normal cells. A20, under the control of NF-κB/RelA, limits apoptosis induced by TRAIL in pancreatic cancer cells. This suggests that it contributes to the resistance of pancreatic cancer cells to TRAIL-mediated apoptosis, highlighting its role in cell survival mechanisms [40].

In addition to breast, melanoma, gastric, and colorectal cancers, A20 has been implicated in the progression of various other malignancies, including endometrial cancer, adrenocortical carcinoma, cholangiocarcinoma, bladder cancer, esophageal carcinoma, and thyroid cancer [41,42,43,44,45,46,47]. In nasopharyngeal carcinoma, A20 is a direct target of miR-19b-3p. Knockdown of A20 reduces sensitivity to irradiation, whereas upregulation of A20 expression reverses the inhibitory effects of miR-19b-3p on cell radio-sensitivity [48] (Figure 1D). These findings show the role of A20 as an oncogene in various cancer types. However, further understanding is necessary to highlight the multifaceted roles of A20 in cancer biology (Table 1).

## 4. Tumor-Suppressive Functions of A20 in Cancer

Chronic inflammation is a well-known risk factor for cancer development, as it generates a microenvironment for tumor initiation and progression. Through its potent anti-inflammatory actions, A20 mitigates this risk by regulating key signaling pathways, such as NF-κB, and maintaining immune homeostasis. Dysfunctional A20 disrupts this immune balance, leading to persistent inflammation and the establishment of a pro-tumorigenic environment [8]. By inhibiting excessive NF-κB activation, A20 prevents uncontrolled cell proliferation and survival, which are hallmarks of cancer [8].

A20 also exerts tumor-suppressive effects by promoting apoptosis in cells with damaged DNA or excessive proliferative signals. This eliminates cells at risk for malignant transformation. Loss-of-function mutations or deletions in the A20 gene have been observed in various cancers, including lymphomas, multiple myeloma, and solid tumors, such as hepatocellular carcinoma (HCC), non-small-cell lung cancer (NSCLC), and pancreatic cancer [9].

The tumor-suppressive role of A20 is particularly evident in lymphomas, where somatic mutations are observed in up to 30% of patients with B-cell lymphomas, such as Hodgkin’s and non-Hodgkin lymphomas and mucosa-associated tissue (MALT) lymphoma [49,50,51]. Functional studies have shown that A20 overexpression induces apoptosis in lymphoma cell lines, whereas the silencing of A20 is associated with resistance to apoptosis and enhanced clonogenicity. These findings suggest that A20 plays an important role in lymphomagenesis by regulating NF-κB signaling [49,50,51].

The loss of A20 leads to constitutive activation of NF-κB, which not only promotes tumor growth but also contributes to chemotherapy resistance in B-cell lymphomas [52]. Drug resistance mediated by A20 depletion is associated with enhanced survival signals provided by NF-κB, which renders cancer cells less susceptible to treatment-induced apoptosis. Despite its well-established role in promoting lymphomagenesis, A20 mutations may have therapeutic implications. For example, patients with A20-mutant B-cell lymphomas exhibit differential responses to certain therapies, such as Bruton’s tyrosine kinase (BTK) inhibitors [53]. This indicates that A20 mutations may improve survival in patients treated with BTK inhibitors, which highlights the complex role of A20 in lymphoma treatment and the necessity for personalized therapeutic approaches. Interestingly, in chronic lymphocytic leukemia (CLL), increased NF-κB activity is observed without genetic alterations in the A20 gene, such as mutations or promoter methylation. This suggests that other mechanisms may be at play in maintaining NF-κB activation in CLL [54]. Similarly, A20 mutations are rare in T-cell acute lymphoblastic leukemia (T-ALL) [55]. These findings indicate the need for further studies to establish the role of A20 in different lymphoma subtypes.

In HCC, A20 expression is inversely correlated with tumor size, suggesting its role as a tumor suppressor [56]. Mouse models with hepatocyte-specific deletion of A20 also develop severe liver inflammation, fibrosis, and eventually HCC [57]. This phenotype is attributed to the increased sensitivity of A20-deficient hepatocytes to inflammatory cytokines, which leads to increased cell death and a subsequent pro-tumorigenic environment. By inhibiting key signaling pathways, such as NF-κB and RIPK1-mediated necroptosis, A20 plays a role in maintaining liver homeostasis and prevents the transition from chronic inflammation to cancer (Figure 2). Other in vitro studies also support this tumor-suppressive role, showing that A20 restricts the proliferation, motility, and metastasis of HCC cells [56,58]. A20 may also regulate various signaling pathways and processes, including NF-κB, RAC1, FAK signaling, and glycolysis metabolism [58,59]. Although these studies demonstrate the involvement of A20 in these mechanisms, the specific pathways regulated by A20 remain to be fully elucidated.

In NSCLC, the role of NF-κB has been extensively studied using mouse models, revealing its critical involvement in tumor development and progression [60]. For example, inhibition of the NF-κB pathway, such as IKKα or IKKβ, reduces lung inflammation and tumor growth and improves survival [61,62]. In A20-deficient models with oncogenic K-Ras activation and p53 loss, enhanced NF-κB activity was associated with increased tumor development, whereas inhibiting NF-κB reduced tumor formation [63]. Another study demonstrated that NF-κB promotes lung inflammation and tumor growth by activating immune cells, such as tumor-associated macrophages, and recruiting regulatory T cells, to generate a pro-tumorigenic inflammatory microenvironment [64]. Moreover, certain miRNAs promote NSCLC progression by regulating A20 expression in NSCLC. miR-29b leads to the activation of NF-κB signaling by targeting A20 to confer apoptosis resistance in NSCLC cells [65]. In addition, miR-605-5p suppresses A20 expression, facilitating cell invasion and proliferation in NSCLC [66] (Figure 2). These results suggest that A20 plays a tumor-suppressing role in NSCLC as a NF-κB negative regulator.

A20 down-regulation impairs CD8^+^ T cell-mediated immune surveillance in lung cancer patients and mouse models [67]. The mechanism involves increased sensitivity to interferon-γ (IFN-γ) signaling resulting from the hyperactivation of TANK-binding kinase 1 (TBK1) and increased expression and activation of STAT1. This leads to elevated PD-L1 levels, which facilitates tumor immune escape. Moreover, immune checkpoint blockade (ICB) treatments are particularly effective in mice with A20-deficient lung tumors, suggesting that targeting the TBK1-STAT1-PD-L1 axis enhances ICB therapy in lung adenocarcinoma patients [67].

A20 also exhibits tumor-suppressive properties in pancreatic cancer. A20 expression levels were shown to be lower in pancreatic cancer tissues compared with normal tissues, regardless of patient age or gender [68], and were significantly associated with aggressive tumor behavior. Furthermore, as a significant prognostic marker in patients with pancreatic ductal adenocarcinoma (PDAC), A20 expression levels were associated with key clinical features, such as TNM stage, tumor differentiation, and patient survival rates. This provides valuable insight for predicting patient outcomes and guiding treatment strategies [69]. A20 is not only a prognostic marker, but also a potential therapeutic target in PDAC.

A20 may also influence chemotherapy resistance. Resistance to gemcitabine, a first-line chemotherapeutic agent for PDAC, remains a major challenge in the treatment of pancreatic cancer. For example, miR-125a reduces gemcitabine sensitivity in pancreatic cancer cells by directly targeting A20. Overexpression of miR-125a results in the downregulation of A20 and increased chemoresistance. Conversely, restoring A20 expression enhances cellular sensitivity to gemcitabine, indicating its potential role in overcoming drug resistance [70] (Figure 2).

Although studies have identified these pathways as potential targets, the precise molecular mechanisms through which A20 modulates these processes remain incompletely understood, particularly in pancreatic cancer development (Table 2).

## 5. Association Between A20 and Autophagy

As a tumor regulatory pathway, autophagy has gained attention as a type of metabolic alteration. Autophagy is a lysosome-dependent degradation pathway that is essential for maintaining cellular homeostasis. It is responsible for clearing damaged organelles, misfolded proteins, and intracellular debris and plays an important role in various biological processes, including organismal development, immune system regulation, and energy homeostasis [71,72,73,74]. Dysregulation of autophagy has been implicated in multiple diseases, including cancer, where it exerts dual roles depending on the context [74].

In the early stage of cancer, autophagy generally acts as a tumor suppressor by reducing genomic instability and clearing potential oncogenic factors [75]. Studies on autophagy-related genes (ATGs) support this tumor-suppressive role. For example, the heterozygous deletion of the BECN1 gene, encoding Beclin1, results in spontaneous tumorigenesis in mice [76,77]. Beclin1 is a key regulator of cell survival, functioning at the intersection of autophagy and apoptosis by dynamically interacting with different protein complexes. It promotes autophagy through association with the Vps34–Atg14 complex, while its binding to anti-apoptotic Bcl-2 family members suppresses cell death pathways [78]. Due to its dual role in autophagy and apoptosis, systemic Beclin1 knockout in mice leads to early embryonic lethality (by E7.5), in contrast to other autophagy gene knockouts like Atg5 or Atg7, which cause neonatal lethality due to impaired nutrient mobilization [79]. This suggests that Beclin1 plays a critical role in early development by participating in cellular processes beyond autophagy, which may also contribute to its tumor-suppressive function. Moreover, in human cancers, mutations in autophagy-related genes such as ATG2, ATG5, ATG9, and ATG12 have been observed in colorectal and gastric cancer, implying a tumor-suppressive role for autophagy [80]. However, direct mechanistic evidence from in vivo cancer models remains limited.

Conversely, in established tumors, autophagy often supports cancer cell survival under stress conditions such as nutrient deprivation, hypoxia, and therapeutic interventions. This adaptation promotes tumor growth and therapy resistance [74,81]. For example, studies using a genetically engineered mouse model with conditional deletion of Atg5 or Atg7 resulted in significant suppression of tumor growth, indicating the tumor-promoting role of autophagy, which was associated with impaired glucose homeostasis in lung tumors [82]. In pancreatic ductal adenocarcinoma (PDAC), autophagy promotes tumor survival by recycling cellular components into essential metabolic substrates under nutrient-limited conditions, thus enabling sustained proliferation and resistance to apoptosis through cancer-autonomous and non-autonomous mechanisms [83].

Recently, A20 has been reported to be closely associated with autophagy regulation across various cellular contexts, including immune cells. As an essential regulator at the interface of inflammation and immune responses, A20 has emerged as a critical modulator of autophagic processes through its interaction with ubiquitin-mediated signaling pathways [84]. This positions A20 as a molecular bridge between inflammatory regulation and cellular homeostasis via autophagy.

Mechanistically, A20 modulates autophagy through its interaction with core autophagy components, particularly those regulated by ubiquitination. For example, Beclin-1 is an essential initiator of autophagosome formation that undergoes K63-linked ubiquitination by TRAF6, which activates autophagy. As an enzyme, A20 removes ubiquitin chains from Beclin-1, thereby inhibiting autophagy in response to Toll-like Receptor 4 (TLR4) signaling [85]. This balance between TRAF6-mediated ubiquitination and A20-mediated deubiquitination of Beclin-1 is a critical factor in regulating TLR4-mediated autophagy.

Beyond its role in Beclin-1 regulation, A20 also attenuates selective autophagy processes by targeting specific signaling components. Under conditions of A20 depletion, TRIF, a downstream adaptor of TLR signaling, is degraded through autophagy, which is mediated by the selective autophagy receptor NDP52. This degradation suppresses inflammatory responses and illustrates how A20 indirectly regulates immune signaling via autophagy modulation [86]. These results suggest a dual role for A20 in regulating both inflammatory pathways and autophagic activity. Such findings also imply that A20 plays a role not only in controlling inflammatory pathways but also in shaping the autophagic landscape within tumors.

The involvement of A20 in autophagy extends to immune regulation, where it modulates inflammatory responses. The mTOR pathway is a central regulator of autophagy, and its inhibition activates autophagy [87]. A20 interacts with components of the mTOR signaling pathway, thereby affecting autophagy. For example, A20 interacts with the mTOR complex, and A20-deficient CD4^+^ T cells show enhanced ubiquitination of this complex and mTOR activity. A20 restricts mTOR signaling and promotes autophagy, which provides insight into how mTOR and autophagy regulate survival in CD4^+^ T cells [88]. In other immune cells, A20 prevents excessive inflammation by stabilizing key signaling components, such as DEPTOR, an mTOR suppressor. This enhances autophagic activity, thereby suppressing inflammasome activation and reducing chronic inflammation associated with autoimmune diseases [89]. This mechanism highlights the ability of A20 to balance autophagy and mTOR activity associated with inflammation in immune cells, thereby preventing aberrant immune responses.

Conversely, autophagy enhances inflammation by sequestering A20 in some contexts. For example, p62, an autophagy adaptor protein, binds to A20 and targets it for sequestration within the autophagosome. The sequestration of A20 reduces its availability to inhibit NF-κB signaling, thereby promoting innate immune responses. This mechanism has been implicated in macrophage activation and may contribute to the pro-inflammatory environment observed in some cancers [90].

A20 acts as a negative regulator of the NF-κB pathway, which influences autophagy by regulating autophagy-related gene expression [91,92,93]. A20 also supports cell survival in certain contexts by enhancing autophagic activity. A20 interacts with the WD40 domain of ATG16L1, a core autophagy component, to modulate autophagy dynamics and NF-κB signaling. This interaction maintains a balance between inflammation and autophagy, allowing the cells to survive under stress conditions [91]. A20 interacts with an autophagy receptor, such as an inhibitor of nuclear factor kappa B (NF-κB)-1 (ABIN-1), a polyubiquitin-binding protein that functions as a signal-induced autophagy receptor. This interaction attenuates NF-κB-mediated inflammation and cell death. The dual functionality of A20 highlights its role in linking autophagy with inflammatory signaling pathways [92].

Given the role of A20-mediated immune signaling in shaping the tumor microenvironment and immune responses, the ability of A20 to suppress autophagy in this context may contribute to tumor-associated immune evasion.

Moreover, the interplay between A20 and autophagy may suggest a significant implication for cancer biology. A20-mediated inhibition of NF-κB can thus modulate autophagic activity, thereby affecting cancer cell survival and growth [91,92,93]. This regulation may either enhance or inhibit autophagy to affect tumor dynamics. A20 influences the balance between cell growth and autophagy, with implications for cancer progression. A20-mediated autophagy prevents cancer by eliminating damaged organelles and proteins, which reduces genomic instability and the potential for tumorigenesis.

A20 also plays an important role in regulating autophagy in cancer. Its interaction with autophagic processes influences cancer cell survival, proliferation, and response to treatment. A recent study found that A20 expression prevents diffuse large B-cell lymphoma (DLBCL) cell proliferation and migration by inducing TLR4/MyD88/NF-κB pathway-mediated autophagy in vitro. This suggests that A20 may act as a tumor suppressor by modulating the autophagic pathways in lymphoma [94].

However, A20-mediated modulation of autophagy impacts cancer cell survival. It plays an important role in modulating sensitivity to TRAIL-induced apoptosis by influencing autophagy-related survival mechanisms. In pancreatic cancer cells, A20 expression is associated with resistance to TRAIL-induced cell death, which is linked to low levels of p62, a key autophagy adaptor protein [40]. A20 downregulation in these cells increases sensitivity to TRAIL-induced apoptosis, which is restored by p62 knockdown.

These findings underscore the importance of understanding the context-specific roles of A20 in the regulation of autophagy in cancer. Although multiple observations have shown the relationship between A20 and autophagy under specific biological conditions, further studies are needed to fully understand the molecular function of A20 as an autophagy modulator and its functional relationship with cancer progression. Given that dysregulated autophagy contributes to cancer cell survival and resistance to therapy, further elucidation of the role of A20 in autophagy regulation may offer novel insights into its potential as a therapeutic target in cancer.

## 6. Conclusions and Future Perspectives

A20 plays a multifaceted role in regulating autophagy, affecting cancer cell survival, proliferation, and immune evasion. Although A20 is primarily known as an oncogene, there are contexts in which it may act as a tumor suppressor. This dual role underscores the complexity of its function in cancer biology. Understanding the specific conditions and mechanisms through which A20 contributes to tumorigenesis is important for developing effective cancer therapies (Table 1 and Table 2).

Furthermore, autophagy is vital for maintaining cell homeostasis and preventing cancer initiation. Similarly to the role of A20 in cancer, the role of autophagy in established tumors is also complex, often supporting cancer cell survival under stress conditions.

The ability of A20 to modulate autophagy-related processes through ubiquitin editing and its interactions with key autophagy components underscores its importance in maintaining cellular homeostasis. Recent studies have identified A20 as a significant regulator of autophagy through its inhibition of the NF-κB pathway and its interaction with the mTOR signaling pathway, thereby affecting cancer cell survival and growth. However, the dual role of A20 as both a suppressor and promoter of autophagy in cancer reflects the complexity of its regulation. Depending on the context, A20-mediated autophagy modulation can either suppress or promote tumor dynamics. Therefore, further studies are needed to fully understand its function as an autophagy modulator in tumor development in various contexts of cancers based on the molecular evidence observed in the immune system.

Finally, studies are needed to elucidate the precise molecular mechanisms underlying the interplay between A20 and autophagy, particularly in the context of cancer. Determining how A20-mediated autophagy influences tumor dynamics, drug resistance, and immune modulation may provide novel insights for therapeutic strategies targeting autophagy and inflammatory signaling in cancer.

## Figures and Tables

**Figure 1 cells-14-00544-f001:**
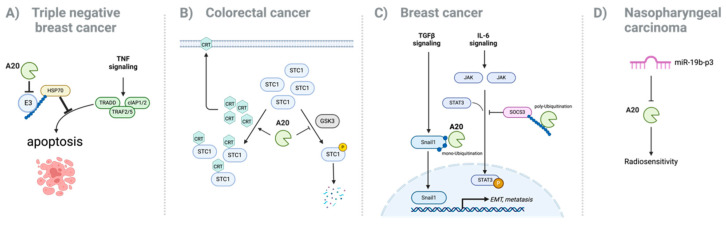
Oncogenic role of A20. A20 plays an oncogenic role in various tumors through various mechanisms. (**A**) Resisting cell death: A20 inhibits the E3 ligase responsible for degrading HSP70—an important regulator of anti-apoptotic signaling—thereby enhancing the HSP70-mediated anti-apoptosis pathway in triple-negative breast cancer. (**B**) Avoiding immune destruction: Under conditions of high A20 expression in colorectal cancer, STC1 binds to calreticulin (CRT), preventing its translocation to the cell surface and thereby suppressing the “engulf me” signal to immune cells, including macrophages. (**C**) Activating invasion and metastasis: In breast cancer, A20 inhibits GSK3-mediated phosphorylation of Snail1 and stabilizes Snail1 by monoubiquitinating it, which leads to the induction of the epithelial–mesenchymal transition (EMT) phenotype. Furthermore, A20 enhances EMT by promoting STAT3 signaling through SOCS3 degradation. (**D**) Therapy resistance: Repressing A20 by miR-19b-p3 increases tumor cell radiosensitivity in nasopharyngeal carcinoma.

**Figure 2 cells-14-00544-f002:**
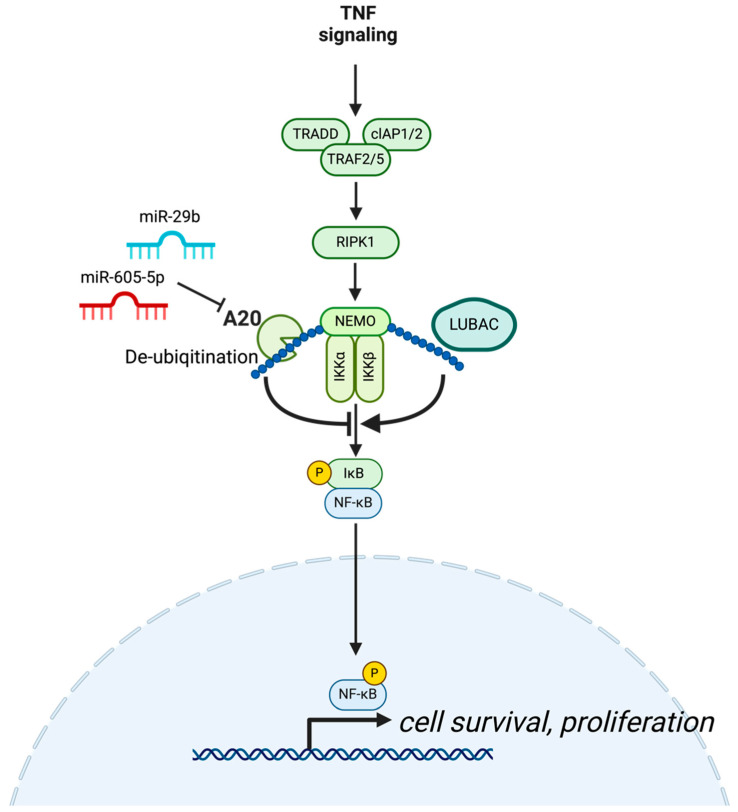
Tumor-suppressive role of A20. A20 serves as a tumor suppressor in cancer cells by removing M1-linked ubiquitin chains from NEMO, thereby inhibiting NF-κB signaling involved in cell survival.

**Table 1 cells-14-00544-t001:** Oncogenic role of A20 in cancers.

Tumor Type	A20 Gene Alteration	Mechanism	References
breast cancer	increased	Suppressing TNF-α-induced apoptosis	[25,26]
		Enhancing metastasis by multi-monoubiquitination of Snail1	[25]
triple-negative breast cancer	overexpression	Activating the HSP70-mediated anti-apoptotic pathway	[26]
luminal (ER+) breast cancer	overexpression	Promoting EMT, increasing metastatic potential via Stat3 signaling, and recruiting granulocytic MDSCs	[26]
melanoma	increased	Enhancing cell proliferation, survival, and metastasis by stabilizing and activating key components of the Akt pathway	[29]
	increased	Promoting immune evasion by regulating PD-L1 expression and decreasing the infiltration and activity of CD8 T cells	[10,11]
gastric cancer	hypomethylation of A20 promoter	Not specifically stated	[31]
	increased	Under Helicobacter pylori infection, inhibiting the NF-κB pathway and reducing expression level of anti-apoptotic genes	[32]
colorectal cancer	genetic and epigenetic alteration	Not specifically stated	[33,34,35,36]
	increased	Promoting tumor growth and immune evasion through a STC1-CRT-dependent pathway by reducing CRT on the tumor cell surface	[37]
pancreatic cancer	Not stated	Contributing to the resistance of pancreatic cancer cells to TRAIL-mediated apoptosis	[38]
endometrial cancer	increased	Correlation of A20 expression with clinical parameters	[39]
adrenocortical carcinoma	increased	Correlation with therapeutic responsiveness	[40]
cholangiocarcinoma	increased	Correlations between A20 expression and patient outcomes	[41]
bladder cancer	increased	Not specifically stated	[42]
esophageal carcinoma	increased	Correlation of A20 expression with patient survival data	[43,44]
thyroid cancer	increased	Genomic analysis for mutation identification	[45]
nasopharyngeal carcinoma	increased	Reducing the inhibitory effects of mir-19b-3p on cell radiosensitivity	[46]

**Table 2 cells-14-00544-t002:** Tumor-suppressive role of A20 in cancers.

Tumor Type	A20 Gene Alteration	Mechanism	Reference
B-cell lymphoma	overexpression	Inducing apoptosis	[47,48,49]
	silencing	Enhancing resistance to apoptosis and clonogenicity	[47,48,49]
	A20 depletion	Contributing to drug resistance and enhancing survival signals provided by NF-κB in cancer cells	[50]
	A20 mutation	Improving cancer cell survival in patients treated with BTK inhibitors	[51]
HCC	decreased	Negative correlation between A20 expression level and tumor size	[54]
	A20 depletion	Developing severe liver inflammation, fibrosis, and HCC	[56]
	A20 depletion	Inhibiting NF-κB-RIPK1-mediated necroptosis	[54,56]
	A20 depletion	Modulating NF-κB, RAC1, FAC signaling, and glycolysis metabolism	[56,57]
non-small-cell lung cancer	A20 depletion	Enhancing NF-κB activity linked to increase in tumor development	[61]
	A20 depletion	By targeting A20, miR-29b activates NF-κB signaling, thereby conferring resistance to apoptosis	[63]
	A20 depletion	miR-605-5p suppresses A20 expression, thereby promoting tumor cell invasion and proliferation	[64]
	A20 depletion	Impairing CD8^+^ T cell-mediated immune surveillance	[65]
		Hyperactivation of TBK1, along with increased expression and activation of STAT1, enhances sensitivity to IFN-γ and elevates PD-L1 levels, ultimately facilitating immune evasion	
pancreatic cancer	decreased	A20 is suggested as a new biomarker	[66,67]
	A20 depletion	miR-125a reduces gemcitabine sensitivity by directly targeting A20	[68]

## Data Availability

No new data were created or analyzed in this study. Data sharing is not applicable to this paper.

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
