# Peer review of "The Role of A20 in Cancer: Friend or Foe?"

_cells, 2025, doi:10.3390/cells14070544_

Round 1

Reviewer 1 Report

Comments and Suggestions for Authors

The manuscript of Jinju Lee, and Heesun Cheong is a literature review on the dual role of A20/TNFAIP3 in cancer. The authors examine the paradoxical role of A20 in cancer by promoting tumor progression or on the contrary by preventing tumorigenesis in different forms of cancers. Associated to its key role in regulating inflammatory signaling and its interaction with autophagy, the authors provide insight how its dysregulation and its associated pathway can interfere with or drive cancer development. The figures are well illustrated and tables are well completed. Overall, I found this study very interesting. However, the manuscript requires major revisions.

Major Revisions:

  • Figure 1 and 2 can be inverted. The authors should indicate their new figure 1 (ex. Fig. 2) in the chapter 2. The authors should change the title of the figure in line with inflammation.
  • Line 90: Figure 1 should become Figure 2.
  • Line 153: The authors should indicate the Table 1 at this level not in the conclusion.
  • Lines 242-250: The paragraph devoted to A20 in TRAIL resistance should be moved to the chapter 3 on the oncogenic role of A20.
  • Line 250: The authors should indicate the Table 2 at this level and not in the conclusion.
  • Lines 286-291: The paragraph should be deleted as it is redundant with the one on lines 292-298.
  • Table 1 and Table 2 should be improved because some lines are empty or not aligned with their reference.
  • Abbreviations should be more complete.

Minor Revisions:

  • Line 17: There is a double point.
  • Line 25: correct ‘ac-tivity’ by activity.
  • Line 27: there is a double space between domain and with.
Comments on the Quality of English Language

The English is fine but but it can be improved.

Author Response

 We appreciate your suggestion and comment. We agree with your valuable comment. Therefore, we revised our manuscript significantly following your suggestions.

Major Revisions:

  • Figure 1 and 2 can be inverted. The authors should indicate their new figure 1 (ex. Fig. 2) in the chapter 2. The authors should change the title of the figure in line with inflammation.

(Answer) Thank you for your comment. In this review, we have more focused on the oncogenic or tumor suppressing role of A20, not covered its role on inflammation deeply here.

  • Line 90: Figure 1 should become Figure 2.

(Answer) We are not able to understand your comment. We only explained the oncogenic role of A20 in figure 1 and tumor suppressing role in figure 2, not point out its role on inflammation in this review.

  •  
  • Line 153: The authors should indicate the Table 1 at this level not in the conclusion.

(Answers) Agree with your opinion. We revised to indicate tables and figures on the text. As reviewer’s mention, we indicated the Table 1 in section 3.

  • Lines 242-250: The paragraph devoted to A20 in TRAIL resistance should be moved to the chapter 3 on the oncogenic role of A20.

(Answers) We apologize our mistake to make confusion. We move the paragraph (Line 242-250) about A20 in TRAIL resistance to the section 3 to explain the oncogenic role of A20 in pancreatic cancer. Also, I removed this mechanism in the figure 2.

  • Line 250: The authors should indicate the Table 2 at this level and not in the conclusion.
  • (Answers) Agree with your opinion. We revised to indicate tables and figures on the text. As reviewer’s mention, we indicated the Table 2 in section 4.

  • Lines 286-291: The paragraph should be deleted as it is redundant with the one on lines 292-298.

(Answers) We apologize our mistake to provide repetitive paragraph (Line 288_291), which is removed.

  • Table 1 and Table 2 should be improved because some lines are empty or not aligned with their reference.

(Answers) We revised Table 1& Table 2 to remove empty lines and align with their references well. We aligned the lines with their reference. Additionally, we filled the blank to “Not specifically stated” if it is not clearly stated in their references.

  • Abbreviations should be more complete.

(Answers) We have added abbreviations more completely

Minor Revisions:

  • Line 17: There is a double point.
  • Line 25: correct ‘ac-tivity’ by activity.
  • Line 27: there is a double space between domain and with.

(Answers) We apologize our mistakes during formatting our manuscript and we all revised the reviewer’s comments.

Reviewer 2 Report

Comments and Suggestions for Authors

This review describes the dual role of A20, an ubiquitin-editing enzyme, that acts as an oncogene or tumor suppressor depending on the type of cancer. Even in the same type of cancer, its effects may differ depending on other factors. In principle, this review is well written and of interest to a broader audience. By contrast, section 5 about autophagy differs from the other parts because this section is not very well structured and it includes some repetitive but not very informative statements. There are many short paragraphs without context. Thus, section 5 has to be thoroughly revised before publication. Here are some examples about the inconsistencies in part 5:

  • Paragraph lines 258-264: The different effects of BECN1 and ATG5 deletion need a better description and the function of BECN1 has to be described first and not later in paragraph 286-290.
  • Paragraph lines 282-285: This paragraph is not well integrated into the text between two paragraphs about A20.
  • The two paragraphs in lines 286-290 and 292-298 are repetitive and they are containing the same information and almost the same sentences.
  • Paragraph lines 335-340: The reference is missing in this paragraph.

Some other parts of the review are also not clear to me:

  • Is A20 a E3 ubiquitin ligase for K63-linked polyubiquitin (line 26) or is A20 deubiquitinating by removing K63-linked polyubiquitin chains from signaling intermediates such as RIP (lines 62-65)?
  • Figure 1: This figure has to be rearranged because it is not clear which parts of the figure belong together. The connections between A, B, C, D and the schemes are missing. Please revise the figure in way to make it more logical.

Author Response

 We appreciate your suggestions and comments. We agree with your valuable comment. As a long-time autophagy researcher, I would like to introduce the new molecular role of A20 on cancer progression, which is mediated by regulating autophagy. Following the reviewer’s suggestion, we reconstructed section 5, including more sentences for logical connection to prior sections, and deleted the repetitive sentences.

  • Paragraph lines 258-264: The different effects of BECN1 and ATG5 deletion need a better description and the function of BECN1 has to be described first and not later in paragraph 286-290.

(Answers) As per the reviewer’s suggestion, we described the general functions of BECN1 and ATG5 in an earlier paragraph (Line 271-273). However, the genetic effect of BECN and ATG5 on tumorigenesis just emphasized the tumor suppressing function rather than their molecular functions

  • Paragraph lines 282-285: This paragraph is not well integrated into the text between two paragraphs about A20.

(Answers) We revised this paragraph to be more integrated into this chapter, which removed some of the sentences not and added sentences typed RED. We deleted that paragraph( line 284-287), which was not well integrated, and instead revised it to be better connected to the prior and following paragraphs (Red typed)

  • The two paragraphs in lines 286-290 and 292-298 are repetitive and they are containing the same information and almost the same sentences.

(Answers) We apologize for the mistake of providing a repetitive paragraph (Line 288_291), which is removed.

  • Paragraph lines 335-340: The reference is missing in this paragraph.

(Answers) We added the references (87-89).

Some other parts of the review are also not clear to me:

  • Is A20 a E3 ubiquitin ligase for K63-linked polyubiquitin (line 26) or is A20 deubiquitinating by removing K63-linked polyubiquitin chains from signaling intermediates such as RIP (lines 62-65)?

(Answers) We delineated that because A20 contains dual domains-E3 and DUB in one molecule, A20 has different functions to target distinct substrates.

  • Figure 1: This figure has to be rearranged because it is not clear which parts of the figure belong together. The connections between A, B, C, D and the schemes are missing. Please revise the figure in way to make it more logical.

(Answers)

This figure shows various molecular functions of A20 on tumorigenesis through A,B,C,D distinctly, in different types of cancer. Each parts would not be connected. Not to be confused, we revised to mention cancer types related to each mechanism in the figure1 legend. Also, we indicated figure 1A,B,C,D on the text separately.

Round 2

Reviewer 1 Report

Comments and Suggestions for Authors

The authors have improved their manuscript and have addressed my comments.

Author Response

We appreciate your suggestion and comment.

Reviewer 2 Report

Comments and Suggestions for Authors

Unfortunately the manuscript was not revised according to my suggestions and the response of the authors is difficult to understand. Therefore, the manuscript should be revised accordingly before publication.

Author Response

We appreciate your suggestion and comment.

This review describes the dual role of A20, an ubiquitin-editing enzyme, that acts as an oncogene or tumor suppressor depending on the type of cancer. Even in the same type of cancer, its effects may differ depending on other factors. In principle, this review is well written and of interest to a broader audience. By contrast, section 5 about autophagy differs from the other parts because this section is not very well structured and it includes some repetitive but not very informative statements. There are many short paragraphs without context. Thus, section 5 has to be thoroughly revised before publication. Here are some examples about the inconsistencies in part 5:

Answers: We sincerely appreciate your thoughtful comments and suggestions. We fully agree with your observation regarding Section 5. As a long-time researcher in the field of autophagy, our intention was to introduce the emerging molecular role of A20 in cancer progression through its regulation of autophagy. In response to your recommendation, we have thoroughly revised Section 5 (All changes; yellow marked) to improve clarity and coherence. Specifically, we enhanced the logical flow by strengthening the connection with preceding sections and removed redundant sentences to avoid repetition.

  • Paragraph lines 258-264: The different effects of BECN1 and ATG5 deletion need a better description and the function of BECN1 has to be described first and not later in paragraph 286-290.

Answers:Following the reviewer’s comment, we reconstructed the section 5.

In Section 5, we introduced the involvement of A20 in the regulation of autophagy. we briefly described the role of autophagy in cancer and introduced Beclin-1 as a key molecule in autophagosome formation. As per the reviewer’s suggestion, we decided to remove the effect of ATG5 in tumor suppressing role (77) to avoid the readers’ confusion. Instead, we described more the additional functions of BECN1 in addition to autophagy (Line 265-273 added), suggesting that Beclin1 might have different function in cancer progression.

Then, in lines 289294, we added more connecting sentences, and in lines 295300, we elaborated on the mechanism of Beclin-1-mediated autophagy to explain how A20, as a deubiquitinating enzyme, regulates this process. Accordingly, we believe this paragraph is appropriately placed within the overall structure of the manuscript

  • Paragraph lines 282-285: This paragraph is not well integrated into the text between two paragraphs about A20.

Answer: We have added a transitional sentence to better integrate the two paragraphs(yellow marked). and improve the logical flow of the section

  • The two paragraphs in lines 286-290 and 292-298 are repetitive and they are containing the same information and almost the same sentences.

Answers; We apologize for the mistake of providing a repetitive paragraph, which is removed. 

  • Paragraph lines 335-340: The reference is missing in this paragraph.

Answers: We added the references (91-93).

Some other parts of the review are also not clear to me:

  • Is A20 a E3 ubiquitin ligase for K63-linked polyubiquitin (line 26) or is A20 deubiquitinating by removing K63-linked polyubiquitin chains from signaling intermediates such as RIP (lines 62-65)?

The answer is not clear.

Answer: Following your suggestion, we revised line 24-25 on introduction and also revised our manuscript (Line60-69) with additional references.

A20 is a dual-function enzyme that acts as a deubiquitinase through its N-terminal OTU domain to remove K63-linked polyubiquitin chains from signaling intermediates such as RIP1, thereby inhibiting NF-κB activation, and also functions as an E3 ubiquitin ligase via its C-terminal zinc finger domains to promote K48-linked polyubiquitination with the aid of ITCH, RNF11, and TAXBP-1, targeting these signaling proteins for proteasomal degradation and further dampening inflammatory responses.

Cell Death and Differentiation (2010) 17, 14–24 

Reference:

  1. Regulation of death receptor signaling by the ubiquitin system, IE Wertz and VM Dixit, Cell Death and Differentiation (2010) 17, 14–24
  2. The ubiquitin-editing enzyme A20 requires RNF11 to downregulate NF-kB signalling Noula Shembade, Kislay Parvatiyar, Nicole S Harhaj and Edward W Harhaj, The EMBO Journal (2009) 28, 513–522
  • Figure 1: This figure has to be rearranged because it is not clear which parts of the figure belong together. The connections between A, B, C, D and the schemes are missing. Please revise the figure in way to make it more logical.

Answer: Figure 1 does not illustrate the connection between panels A–D; instead, it presents various mechanisms observed in different types of cancer. To improve clarity, we organized the figure based on cancer type rather than mechanism.